# Node Oligorecurrence in Prostate Cancer: A Challenge

**DOI:** 10.3390/cancers15164159

**Published:** 2023-08-17

**Authors:** Almudena Zapatero, Antonio José Conde Moreno, Marta Barrado Los Arcos, Diego Aldave

**Affiliations:** 1Health Research Institute, University Hospital La Princesa, 28006 Madrid, Spain; 2Department of Radiation Oncology, La Fe University and Polytechnic Hospital, 46026 Valencia, Spain; conde.ant@gva.es; 3University Hospital of Navarra, 31008 Pamplona, Spain; marta.barrado.losarcos@navarra.es; 4University Clinical Hospital of Valladolid, 47003 Valladolid, Spain; aldavediego@gmail.com

**Keywords:** oligometastases, prostate cancer, PSMA-PET, stereotaxic body radiotherapy (SBRT), oligorecurrent prostate cancer, metastasis-directed therapy, clinical trials

## Abstract

**Simple Summary:**

Nodal oligorecurrence in prostate cancer is currently a topic of extensive research mainly due to its indolent spread pattern and to the remarkable advancements in molecular imaging methods like PSMA-PET. These innovative techniques can identify cancer at lower PSA levels compared to traditional imaging methods. Furthermore, localized ablative radiotherapy treatments like stereotactic body radiotherapy (SBRT) are now accessible, offering enhanced local control and minimal adverse effects. As a result, metastasis-directed therapy with SBRT is becoming an increasingly attractive treatment option for these patients. Despite these advancements, there are still many uncertainties that require further investigation. This review aims to summarize the data supporting this approach, discuss controversies concerning the selection of patients and the best treatment approach, and explore ongoing phase III trials and the future of treating nodal recurrence in prostate cancer.

**Abstract:**

Within the oligometastatic state, oligorecurrent lymph node disease in prostate cancer represents an interesting clinical entity characterized by a relatively indolent biology that makes it unique: it can be treated radically, and its treatment is usually associated with a long period of control and excellent survival. Additionally, it is an emergent situation that we are facing more frequently mainly due to (a) the incorporation into clinical practice of the PSMA-PET that provides strikingly increased superior images in comparison to conventional imaging, with higher sensitivity and specificity; (b) the higher detection rates of bone and node disease with extremely low levels of PSA; and (c) the availability of high-precision technology in radiotherapy treatments with the incorporation of stereotaxic body radiotherapy (SBRT) or stereotaxic ablative radiotherapy (SABR) technology that allows the safe administration of high doses of radiation in a very limited number of fractions with low toxicity and excellent tolerance. This approach of new image-guided patient management is compelling for doctors and patients since it can potentially contribute to improving the clinical outcome. In this work, we discuss the available evidence, areas of debate, and potential future directions concerning the utilization of new imaging-guided SBRT for the treatment of nodal recurrence in prostate cancer.

## 1. Introduction

The actual status of nodal oligorecurrence in prostate cancer (PCa) is an area of active research. Ongoing studies aim to enhance our understanding of its biology in the metastatic process and to devise more efficient treatment approaches. Since no tissue- or blood-based biomarkers for the identification of patients with oligometastatic cancer have already been identified [1], the use of close PSA monitoring after primary treatment together with active incorporation of more sensitive and specific molecular imaging modalities (MIM), such as the PSMA-PET for local and systemic staging, has emerged as a crucial approach for early detection of nodal oligorecurrence [2,3,4]. 

As a result, metastasis-directed therapy (MDT) has become an increasingly important treatment option for these patients, based on reported evidence that suggests that local ablative therapy in general, and stereotactic body radiation therapy (SBRT) specifically, may be very effective in controlling oligometastatic disease [5]. SBRT is a form of real- or near-real-time image guidance radiation therapy that uses high doses of radiation delivered in a precise and targeted method to the tumour, in a very limited number of fractions with low toxicity and excellent tolerance. The initial goal for integrating SBRT into multimodality treatment strategies for nodal oligorecurrence has been to prevent further metastasis development through eradication of nodal disease and/or delay the need for long-term ADT, with the subsequent benefit on the quality of life.

So far, it is still unknown whether the identification, upstaging with PSMA-PET, and subsequent focal treatment of low-volume nodal recurrences improves the survival of these patients. This critical review synthetizes the high-level data supporting the use of biological imaging-guided SBRT as a form of MDT for nodal oligorecurrent disease in hormone-sensitive PCa (HSPCa) in an attempt to address the uncertainties related to the quality and quantity of the benefit of the patient selection and the individualized treatment approach. Lastly, we will discuss the future of nodal recurrence treatment in prostate cancer, including ongoing phase III trials.

## 2. Clinical Evidence and Controversies

### 2.1. Justification and Rationale

The oligorecurrent disease is a clinical scenario usually associated with more favourable outcomes in comparison to synchronous oligometastases, likely due to a more indolent biology and a lymphotropic pattern of recurrence. Clinical evidence suggests that patients with late metastases, defined as occurring more than 2 years after the treatment of the primary tumour, tend to exhibit better survival rates than those with early onset, which implies a slower-growing and less aggressive disease [6]. Multiple studies utilizing MIM have consistently identified the oligonodal presentation as the predominant pattern of recurrence following primary treatment in PCa [7,8]. Additionally, when examining new recurrence patterns after nodal ablative treatment, up to 50% of patients experienced an oligometastatic relapse in neighbouring node areas, with a median time to progression of 19–22 months [9,10].

### 2.2. Summary of Clinical Evidence

Despite the absence of clinical evidence from phase III randomized trials assessing the impact of MDT on clinical outcomes, numerous prospective phase I and II clinical trials conducted over the past decade have evaluated the role of SBRT in the prevention of disease progression in patients with oligometastatic PCa. While these studies did not specifically focus on oligorecurrent nodal disease, it is important to highlight that the nodal site was the prevailing presentation in most of the series (Table 1).

The SABR COMET phase II randomized trial has been the first study to show a significant benefit in overall survival (OS) and progression-free survival (PFS) with SBRT and standard of care (SOC) compared to SOC alone, which is a benefit that persists with longer-term follow-up (8-yr OS: 27.2% vs. 13.6%; 8-yr PFS: 21.3% vs. 0.0) [11]. The drawback of this pivotal trial is that it only included 16 patients with PCa, 14 of whom received SBRT. The findings of the exploratory subgroup analysis that focused on PCa patients have been recently reported [12]. Interestingly, the median PFS for patients in the SOC treatment arm was 11 months, while the median PFS was not reached in the SBRT arm, and the PFS rate remained at 56% at 7 years.

The leading evidence comes from two phase II clinical trials (STOMP and ORIOLE) that randomized asymptomatic HSPCa patients with oligometastatic disease (1–3 lesions) to either MDT (SBRT used in 81% of patients in the STOMP trial and 100% in the ORIOLE trial) or surveillance [13,14]. Despite the small sample size and the short follow-up (FU), both trials observed that MDT as compared with observation, was associated with improved treatment outcomes. SBRT to all visible lesions was able to delay the initiation of ADT and increase the PFS without jeopardizing the quality of life (QoL). The long-term outcomes of pooled data from both trials have been recently reported [15]. With a median follow-up time for the entire group of 52.5 months, MDT remained associated with improved PFS compared to observation (pooled hazard ratio (HR), 0.44; 95% CI, 0.29 to 0.66; *p* value 0.001). Interestingly, the PFS beyond 4 years was 15–20% with SBRT, suggesting that a non-negligible number of patients will benefit from a durable response with MDT. Although further follow-up is needed, these encouraging results indicate that in appropriately selected patients, MDT (and specifically SBRT) without systemic therapy might be a reasonable option in well-informed patients wishing to avoid the side effects of androgen deprivation. 

A post hoc analysis of a masked (blinded) PSMA-PET investigational study in the ORIOLE trial revealed that men who had received treatment for all the lesions identified by PSMA-PET had notably improved PFS (HR 0.26; 95% CI, 0.09–0.76, *p* = 0.006) and distant metastasis-free survival (HR 0.19; 95% CI, 0.07–0.54, *p* < 0.001) than those patients with any untreated lesion (PET positive but conventional imaging negative). These results underscore the relevance of new MIM in determining the potential efficacy of MDT.

Three prospective phase I–II nonrandomized trials (PSMA MRgRT, POPSTAR, and OLI-P) have also studied the impact of MDT in oligorecurrent HSPCa staged by novel MIM [16,17,18]. MRgRT and OLI-P trials used PSMA-PET in the diagnosis, while the POPSTAR study used sodium fluoride (NaF)-PET/CT. ADT was not allowed in PSMA MRgRT and OLI-P trials and was administered in 33% of POPSTAR. Most enrolled patients had less than three lesions. These molecular-screened trials showed that approximately half of the patients achieved a PSA response, resulting in a favourable ADT-free survival rate of around 40–49% at 2 years. A recent meta-analysis of six SBRT studies reported between 2013 and 2020 have also confirmed these data [19,20,21,22].

Despite differences in baseline imaging, inclusion criteria, and primary endpoints, the available evidence supports the existence and clinical relevance of an oligometastatic/oligorecurrent state in PCa and shows that SBRT can effectively control treated lesions and reduce disease progression in a subset of patients with acceptably low toxicity.

Nevertheless, and despite MDT, it is worth noting that a remarkable proportion of patients ranging from 15% to 25% still experienced biochemical progression, suggesting the possible presence of undetectable micrometastatic disease with current MIM technologies. This naturally leads us to the reasonable conclusion that a combined approach of MDT/SBRT with systemic treatment may be required in selected patients.

### 2.3. New Molecular Imaging: Clinical Meaning on Treatment Strategies

The upcoming challenge is to translate the results of recent trials on low-volume metastatic/oligometastatic HSPCa, which have established a new standard of care based on conventional imaging, to the new scenario of molecular imaging-guided MDT therapy. One example of how complex this debate will become is illustrated by the STAMPEDE trial of prostate radiotherapy in low-volume metastatic hormone-sensitive prostate cancer [23] since the immediate issue is whether we should also consider including all “oligomet” and “oligonodes” sites in this scenario.

The growing integration of novel MIM has led to the recognition of a phenomenon known as “lead-time bias,” which has prompted inquiries about the possible predictive importance of molecular diagnosis. In a recent multicentre retrospective study, Sutera et al. [24] explored the clinical and genomic variations between a oligometastatic diagnosis detected through MIM or conventional imaging in 295 patients with HSPCa. The study revealed that patients diagnosed with MIM had fewer TP53 mutations and had better overall survival (OS) than those diagnosed with conventional imaging. The observed difference in overall survival from the time of localized PCa diagnosis as well as from the oligometastases’ occurrence might suggest that MIM oligometastatic PCa could be a more “indolent” disease and not just a result of lead-time bias. However, based on the presented data, it remains uncertain whether PSMA-PET-guided MDT improves outcomes by detecting more indolent diseases or targeting occult diseases.

Although SBRT has been utilized as an approach to delay ADT and avoid the decline in quality of life associated with androgen suppression in what is considered a clinically low-risk scenario, a significant proportion of patients still experience early progression. This suggests the presence of undetected and disseminated diseases that could benefit from a combined systemic approach. Additionally, it is important to note that systemic therapy with ADT and ARPIs is considered the standard of care for metachronous hormone-sensitive low-volume disease [25], and we still do not know whether and how these results might impact the treatment strategy of nodal oligorecurrent PCa. The recently presented data from the EMBARK trial [26] in men with high-risk biochemical recurrence that is nonmetastatic on conventional imaging might have further implications. The findings indicating a noteworthy decrease in metastasis risk when combining ADT and enzalutamide present an intriguing framework to consider within the present MIM nodal recurrent landscape. This is especially true when considering the likelihood that a substantial percentage of the high-risk patients of EMBARK trial might reveal oligometastatic disease if they were to undergo PSMA-PET imaging.

In this complex scenario, the optimal clinical endpoints for MDT trials in PCa remain uncertain, and the use of ADT-free survival or biochemical control should be re-evaluated. It seems reasonable that if the primary goals of SBRT in nodal oligorecurrent PCa are to achieve local control, prevent further metastasis, and delay subsequent systemic treatment escalation in selected patients, we should define our endpoints accordingly. A relevant issue that deserves further investigation is whether metastases-free survival, as measured by conventional imaging, can also act as a proxy for overall survival in patients with hormone-sensitive oligorecurrent PCa detected through molecular imaging.

### 2.4. Tailoring Treatment Approach

DNA biomarkers, genetic and genomic profiling, and ctDNA testing are transforming the management of localized and metastatic disease. They are also expected to play a significant role in tailoring treatments for oligometastatic and oligorecurrent PCa [27]. Despite the limited reported evidence in this scenario, several studies underline the need for novel prognostic and predictive biomarkers and their more than promising role in a personalized treatment approach. 

A sub-analysis was conducted on pooled data from STOMP and ORIOLE trials to assess the effectiveness of a high-risk mutational signature in stratifying outcomes after MDT, which was defined as the presence of pathogenic somatic mutations within ATM, BRCA1/2, Rb1, and TP53. Long-term results showed that MDT benefitted both patients with and without a high-risk mutation, although those individuals without a high-risk mutation treated with MDT had superior outcomes (median PFS 13.4 months versus 7.5 months; HR, 0.53; 95% CI, 0.25 to 1.11; *p* = 0.09) [15]. These findings suggest that selected patients with oligometastatic or oligorecurrent disease lacking a high-risk mutation could potentially de-escalate treatment using MDT alone. Conversely, patients with high-risk mutations should be contemplated for an intensified targeted approach.

RNA expression profiling for genomic classification (GC) has shown potential in identifying aggressive PCa and providing guidance for treatment decisions in localized PCa. A recent systematic review and expert consensus report indicates that GC could not only be helpful as a predictive factor for clinical outcomes after MDT or combined therapy in oligometastatic or oligorecurrent disease but also as a valuable risk stratification biomarker for radiotherapy-tailored treatment [28]. In the oligorecurrent scenario, GC could serve to identify “low-risk” patient candidates for de-escalation with involved nodal SBRT alone from those with “high-risk” features susceptible to “intensification” with extensive pelvic field radiotherapy and focal dose intensification.

The WNT signaling cascade plays a crucial role in regulating multiple cellular processes involving stem cell function, proliferation, survival, and differentiation. Deregulation in WNT signaling has been related to high-risk factors (such as Gleason score and elevated PSA serum levels), disease recurrence, higher metastatic incidence, and reduced overall survival [29]. Within this line of research, a recent multi-institutional retrospective study investigated the clinical implications of genomic alterations in the WNT signaling cascade in a series of 277 metachronous oligometastatic PCa patients. The results showed that WNT pathway mutations were associated with worse OS in oligometastatic hormone-sensitive PCa patients and that outcomes might be improved with MDT [30].

Finally, a combination of genomic signatures and PSMA-PET tumour staging could be of particular interest in oligometastatic and oligorecurrent disease, enabling the identification of patients more suitable for MDT. The study conducted by Sutera et al. [24] provides the first evidence of clinical and biological differences between MIM and conventional imaging identifying oligometastatic HSPCa. This information will probably introduce novel perspectives for future investigations.

However, there are still many open questions concerning the degree to which genomic profiling and molecular imaging can provide supplementary information in relation to outcome prediction or treatment decisions. Xu et al. analyzed the potential of GC scores to predict the presence of occult metastatic disease in 91 patients staged with PSMA-PET [31]. They found a significant association between a higher GC score and pelvic (odds ratio (OR) 1.38 per 0.1 units; *p* = 0.009) or any distant nodal involvement (pelvic or distant; OR 1.40 per 0.1 units; *p* = 0.007). According to the authors, these findings might suggest that patients with higher GC scores could potentially benefit from supplementary nodal imaging and treatment intensification (lymphadenectomy or pelvic nodal irradiation, and/or the addition of systemic agents). 

All these data indicate that proper genomic and biologic signatures could help to select the patients who are most likely to experience a meaningful response to local consolidation and are therefore potential candidates for systemic de-escalation from those patients who may require intensification with novel treatment paradigms. Raising awareness about the relevance of clinical trials is crucial for advancing our understanding and management of this disease.

Meanwhile, we will need to rely on other tumour- or patient-related characteristics to individualize treatment strategies, such as tumour volume (size or number and localization of positive nodes: N1 versus M1a), tumour kinetics (PSA value and doubling time, interval from primary treatment to nodal recurrence), optimal timing of PSMA- PET, risk subgroup at diagnosis, without ever forgetting the patient’s comorbidities, expectations, and preferences. It should be noted that, eventually, the long-term benefit of early detection via PSMA-PET and aggressive treatment may not benefit patients with very indolent biochemical relapse.

Ongoing trials, such as the DART trial (ClinicalTrials.gov identifier: NCT04641078), START-MET trial (ClinicalTrials.gov identifier: NCT05209243), and RAVENS trial (ClinicalTrials.gov identifier: NCT04037358), which combine systemic therapy or radiopharmaceuticals, may help define novel paradigms and shed more light on the role of genetic biomarkers in this population. 

## 3. Combined RT and Systemic Therapy

Although the current SOC at a nodal oligorecurrence typically involves the use of ADT associated or not with ARPIs [23], the prospective studies published to date have mostly evaluated MDT without ADT, with the aim of delaying its onset and improving the quality of life of patients. As previously mentioned, the main phase II trials in oligorecurrence disease have shown a clear benefit in PFS by deferring ADT and its side effects. These results suggest that this strategy is a reasonable option in a subgroup of well-selected patients in whom the main objective is to delay the effects of androgen suppression.

However, the use of MDT with ADT remains poorly studied in this setting, although it could surely be a more effective strategy. Only one phase II trial (SABR-COMET) [11] allowed the combination of both treatments to achieve a significant benefit in OS, with a benefit that persists with longer-term follow-up. These good results are not surprising since ADT has been a mainstay of treatment for advanced prostate cancer, and its combination with radiotherapy has consistently shown good results in other disease scenarios.

In the localized setting, the combination of ADT with prostatectomy has shown disappointing results [32]. Neither surgical complication rates nor oncological endpoints were improved by adding ADT, so the combination remains controversial. On the other hand, ADT in combination with external beam radiotherapy has widely demonstrated, in several clinical randomized prospective trials, improvements in the main oncological endpoints [33,34,35,36,37]. This effect is not only explained by the addition of the antitumoural effects of both treatments but because of the synergistic effect caused by the cellular radiosensitivity that generates androgen suppression in PC cells. 

This synergistic effect has been extensively studied in preclinical models [38,39] that show how the anti-neoangiogenic effect of ADT contributes to normalizing irrigation and oxygenation of the tumour microenvironment making RT more effective [40,41,42,43]. In addition, ADT works as a radiosensitizer by inhibiting the tumour cell’s ability to repair double-stranded DNA damage [42].

But surely the most relevant effect of the combination, in the setting of a nodal oligorecurrence, is the effect of ADT in the distance, that is, outside the radiation field. ADT can eradicate distant micrometastases and, consequently, reduce the risk of distant failure and improve the results in an oligorecurrence. Furthermore, it is hypothesized that ADT can regulate immune responses, improving micrometastases’ control [43,44,45].

This effect is essential to achieve good results in the context of a distant recurrent disease because, even in the age of PSMA-PET, it is hard to identify distant micrometastases. In a curative-intent MDT prospective phase II trial [46] only 22% of patients treated with PSMA-PET-guided MDT achieved a complete response of oligorecurrence after maximal local therapy. Therefore, 78% of these patients presented a subsequent relapse despite good control of visible disease by PSMA-PET. This tells us about the importance of a systemic treatment that covers micrometastases when the disease is not already localized.

Thus, the main ongoing phase III trials use the combination to achieve the best oncological results in a context in which the disease can be well controlled (and potentially curable) with the best available oncological treatment in candidate patients.

Otherwise, in the context of metastatic oligorecurrent disease, in which the SOC is androgen suppression, irradiation of visible macroscopic disease can provide benefits by maintaining the effectiveness of systemic treatment and achieving longer biochemical recurrence-free survival. By acting through different mechanisms, RT manages to minimize androgen-independent clones and delays castration resistance [47]. Additionally, by reducing the overall tumour burden, RT makes hormonal therapy more effective in controlling the disease. Therefore, the addition of SBRT to hormonal treatment promises to be a highly effective strategy that will achieve the best oncological results and will allow the de-escalation of hormonal treatment in certain cases.

There is currently no available clinical evidence from randomized phase III trials assessing the role of ADT in the scenario of node oligorecurrent PCa. However, information from the SBRT-SG 05 prospective multicentre phase II trial provides insight into the subject [48]. This study examined the effectiveness of SBRT and ADT in patients with oligorecurrent hormone-sensitive prostate cancer. The oligorecurrent stage was defined as having less than five bone or lymph node metastases identified by choline PET-CT or/and WB-DWI-MRI. From July 2014 to December 2019, a total of 81 patients from 14 centres were included, with 117 oligometastases treated: 67 in lymph nodes and 50 in bones. After a median follow-up of 41 months, the median distant progression-free survival (DPFS) was 54.2 months (95% CI = 48.2–60.3). There were no reports of toxicity above Grade 3, and the tolerance and toxicity profiles were excellent.

We also have indirect evidence from a scenario that, although it precedes in the natural history of the disease, bears many similarities. This is the case of nonmetastatic biochemical relapse after prostatectomy. In this setting, ADT has shown in three phase III trials of salvage radiotherapy a benefit in reducing the risk of biochemical or clinical progression and death when associated with normo-fractionated radiotherapy [49,50,51]. The population of these studies (and more particularly patients with high-risk features), showed benefits from ADT even though they did not have nodal or metastatic involvement either at diagnosis or at the recurrence. Therefore, it is conceivable that patients with a nodal oligorecurrence who are candidates for MDT will also benefit from adding ADT.

The question at this point will be the optimal duration of ADT. The final results of RADICALS-HD will provide insight into this subject for post-operative patients. However, preliminary findings suggest that long-term ADT (24 months) may offer benefits over short-term ADT (6 months) [52]. Similarly, in the oligorecurrence setting, the MDT can manage to temporarily (or permanently) suspend the ADT in this subgroup of patients with a more indolent biology and who present longer responses to the treatments. This assumption needs to be further investigated in studies such as the EXTEND trial [53], in this phase II randomized basket trial, patients with five or fewer metastases were randomized to HT with or without MDT. A planned ADT break occurred 6 months after enrolment. At a median follow-up of 22.1 months, PFS was improved in the MDT group (*p* < 0.001; HR = 0.25 [95% CI, 0.12 to 0.55]).

## 4. Volume of Treatment and RT Scheme: N1 vs. M1A? Role of Elective Pelvic RT

Oligorecurrent nodal disease represents a special challenge in oligometastatic PCa. The studies conducted on SBRT in this scenario have demonstrated encouraging outcomes [11,13,14,16], and as previously mentioned, more than 50% of patients have lymph node involvement exclusively. A noteworthy observation is that the results of MDT in patients with exclusive nodal involvement were comparatively superior, although the statistical significance was not reached [13]. However, it is important to acknowledge that these studies were not specifically designed to assess these variations. 

Research on the patterns of failure after MDT [13,54,55,56] has shown that most patients with an initial nodal disease experienced a lymphotropic pattern of recurrence, characterized by only a few metastatic lesions and a long interval of disease-free survival (DFS). These characteristics render them appropriate candidates for further rounds of SBRT, offering the potential to extend their DFS in the long run. Conversely, this model of spread also prompts the question of whether elective node radiotherapy (ENRT) in lymph node areas of known risk in conjunction with an extra boost to any PET-positive node, could diminish the likelihood of subsequent pelvic relapses and enhance clinical outcomes compared to SBRT alone (Figure 1).

At present, there is no available evidence from randomized phase II–III trials about the optimal radiotherapy approach in oligorecurrent PCa disease. The most reliable evidence for ENRT is based on retrospective and prospective nonrandomized studies that used different doses and schedules of radiotherapy. 

De Blesser et al. [57] conducted a study that focused on patients with hormone-sensitive nodal oligorecurrent PCa (five or fewer nodes). The authors compared the outcome and toxicity of two metastasis-directed approaches, SBRT alone versus ENRT with or without a boost. ADT was permitted and ranged between 23% in SBRT treatments and 60% in ENRT. The study included 506 patients from 15 different treatment centres, all of whom had regional (N1) and/or distant (M1a) node metastases. The diagnosis was made using either PET/CT (choline: n = 428; PSMA: n = 46; fluorodeoxyglucose: n = 17) or conventional imaging (MRI: n = 5; CT: n = 10). The results revealed that ENRT was linked with fewer nodal recurrences in comparison to SBRT (*p* < 0.001), although with higher toxicity (16% vs. 5%, *p* < 0.01). 

The OLIGOPELVIS-GETUG P07 [58] study is a phase II trial designed to evaluate the effectiveness of high-dose salvage ENRT in combination with 6 months of ADT in patients with oligorecurrent (five or fewer) pelvic node relapses in PCa, which were detected by PET choline imaging. The trial included 67 patients from 15 centres, with approximately 84% having only 1–2 PET-positive nodes. Treatment involved 54 Gy in 1.8 Gy fractions administered to the whole pelvis, with a simultaneous integrated boost of 66 Gy in 2.2 Gy fractions to pathological pelvic lymph nodes. After a median follow-up of 49.4 months, the 2- and 3-year progression-free survival rates were 81% and 58%, respectively. The median progression-free survival was 45.3 months, with genitourinary and gastrointestinal toxicities of Grade 2 or higher occurring in 10% and 2% of patients, respectively, at 2 years.

All these findings, taken together, suggest that ENRT may be more effective than SBRT for managing nodal oligorecurrence. However, it should be noted that an important limitation is the higher use of ADT in ENRT in both De Blesser and GETUG P07 trials. Moreover, the SBRT-SG05 study [48], which evaluated the efficacy of SBRT and ADT, observed a median DPFS of 54.2 months (95% CI = 48.2–60.3), which is not notably distinct from the results seen with ENRT.

In an effort to provide some answers, the PEACE V-STORM [59] phase II randomized trial investigates the potential benefit of metastasis-free survival (MFS) of ENRT with a focal boost as an alternative to nodal SBRT or as an adjuvant following salvage lymph node dissection (sLND) for patients with oligorecurrent nodal PCa relapsing after primary treatment. Although the oncologic outcomes are pending, the first toxicity data indicates that acute grade 2 or higher genitourinary and gastrointestinal toxicity was somewhat higher in the whole pelvic radiotherapy (WPRT) group (13% vs. 8% and 3% vs. 1%, respectively) [60]. In the meantime, we are faced with an ongoing debate about localized treatment (SBRT) versus more extensive radiotherapy (ENRT), along with their combination with ADT and/or ARPIs, and the optimal timing and duration of such treatments. 

A particular area of concern is the treatment approach for patients who experience a recurrence in non-regional nodes. The classification of common iliac nodes (CILNs) as M1a by the AJCC poses a new challenge for both newly diagnosed and relapsed low-volume oligometastatic disease. The current standard of care for these patients is the combination of ADT and androgen receptor pathway inhibitors (ARPIs) without local therapy. However, recent reports have shown that patients treated with definitive radiotherapy and limited ADT have had successful outcomes. 

Choped et al. [61] have recently reported a retrospective comparison of 130 M1a or cN1 PCa patients treated with ADT and radical radiotherapy: WPRT was administered for cN1 disease and extended field WPRT for M1a nodal involvement. The study showed similar outcomes for N1 and M1a in terms of PFS and OS and with little toxicity. The 5-year biochemical-PFS was 77.4% vs. 70.4%, *p* = 0.43, MFS was 86.9% vs. 79.4%, *p* = 0.23, and OS was 92.6 vs. 90.1%, *p* = 0.8, for N1 and M1, respectively. Rich et al. [62] have also presented preliminary results from a small cohort of 34 M1a oligorecurrent patients diagnosed with conventional or PET-based imaging and treated with extended WPRT (45–50 Gy) with an integrated node boost (60–65 Gy in 25 fractions). The study included hormone-sensitive and castration-resistant patients. The 2-year- PFS and OS were 83.4% and 100%, respectively. These promising results may pave the way for prospective studies of imaging-guided intensified SBRT combined strategies for M1a or non-regional node oligorecurrence.

## 5. Trials Ongoing and Future Directions

Currently, the selection criteria for the indication of MDT are based primarily on the number of metastases detected on PSMA-PET [63]. The lack of a biology-based selection criterion may be behind the rapid poly-metastatic progression of some patients, especially those who have not received systemic treatment [15]. Fortunately, there is increasing interest in investigating biomarkers that could help select which patient would benefit from treatment intensification [64].

A crucial goal in the era of PET-CT imaging is to determine whether there is an advantage in selectively irradiating the areas identified on PET-CT or if the pelvic region should be irradiated with an additional boost on enhancing adenopathies. The STORM study has been specifically designed to investigate both strategies, in addition to a 6-month regimen of ADT, using a randomized design. Preliminary findings suggest that there are no noticeable differences in terms of toxicity. However, further evaluation is necessary to assess whether there are any variations in PFS [65]. 

Although the primary focus of MDT in nodal oligorecurrent PCa was initially on delaying ADT (as supported by data from STOMP and ORIOLE trials), clinical evidence from high-risk PCa has established a synergistic effect between ADT and radiation, with a proven impact on overall survival [34,36,37]. Consequently, there is a growing interest in exploring the use of MDT in conjunction with limited “sequences” of ADT and/or newer-generation antiandrogens (using an intermittent approach rather than an indefinite one) to enable longer breaks from systemic treatment and potentially prevent or delay the development of widespread metastases [66]. 

Deek et al. recently presented the results of a multi-institutional cohort study involving 263 patients with oligometastatic prostate cancer who underwent MDT at the 2023 AUA meeting [67]. In this retrospective analysis, the authors observed that the addition of ADT to MDT was associated with an improved time to biochemical progression (HR 0.23; 95% CI 0.16–0.33, *p* < 0.001). We need to await data from several studies, such as the DART trial by Ost et al. (ClinicalTrials.gov identifier: NCT04641078) or the START-MET trial (ClinicalTrials.gov identifier: NCT05209243), to determine the most effective combined strategy of hormonal therapy and SBRT [68] (Table 2).

Conversely, the EXTEND phase II randomized trial was conducted among men diagnosed with oligometastatic PCa to assess whether the inclusion of MDT alongside intermittent hormone therapy yielded improved oncologic outcomes compared to intermittent hormone therapy alone [53]. Following a median follow-up period of 22.0 months, the combined therapy arm demonstrated enhanced progression-free survival (PFS) in comparison to the arm receiving hormone therapy alone. Although this study necessitates confirmation through phase III trials, it presents an intriguing approach that has the potential to positively impact financial toxicity. Thus, it could serve as a compelling option for individuals with indolent oligometastatic PCa since 6 months of intermittent hormone therapy might be inadequate for prolonged control.

A relevant aspect to consider is that ARPi is not the sole systemic treatment under evaluation for the oligometastatic scenario. Numerous trials are incorporating other systemic agents such as immunotherapy, Radium-223, and even Lutetium-PSMA. Additionally, the utilization of various conventional techniques, including CT, bone scintigraphy, or PET/CT PSMA, plays a significant and distinctive role in assessing its final clinical impact. Another consideration is the distinction between continuous and intermittent systemic treatment based on the PSA response. Clearly defining the objectives of these strategies is essential for understanding their purpose and evaluating their effectiveness.

## 6. Conclusions

The available evidence supports the clinical significance of an oligorecurrent nodal state in PCa and shows that molecular imaging-guided SBRT can effectively control treated lesions and decrease disease progression in a subset of patients with minimal side effects. However, a notable proportion of patients will benefit from a combined systemic approach.

By identifying the appropriate genomic and biological characteristics, it will become possible to select patients who are most likely to experience a meaningful response to SBRT consolidation and may thus be considered for systemic de-escalation from patients who may require intensification with innovative treatment strategies. Assessing response adequately, through indicators like PSA levels or the absence of tracer uptake in diagnostic imaging, will also enable us to identify individuals who can temporarily pause systemic treatment, potentially opting for intermittent treatment, thereby improving quality of life while prolonging survival. Enhancing awareness about the importance of enrolling patients in clinical trials is pivotal for advancing in our comprehension and management of this disease.

## Figures and Tables

**Figure 1 cancers-15-04159-f001:**
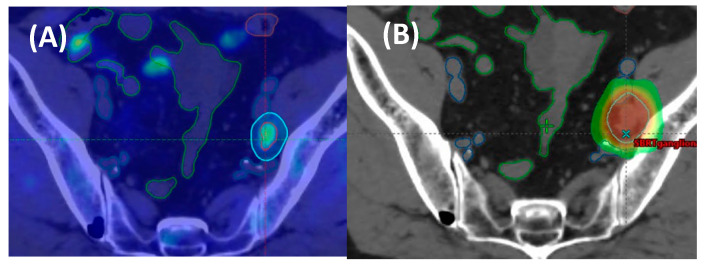
Example of (**A**) a patient with solitary oligorecurrent pelvic nodal relapse detected via PSMA-PET and (**B**) the corresponding imaging matched SBRT plan capture (30 Gy in 3 fractions); (**C**) a patient with multiple (3) oligorecurrent pelvic and low paraaortic nodal disease in PSMA-PET and (**D**) the matched radiotherapy to the extended field pelvic nodes (45 Gy in 25 fractions) with a simultaneous integrated boost to 65 Gy.

**Table 1 cancers-15-04159-t001:** Summary of prospective phase I–II trials of MDT in nodal oligoMET prostate cancer.

Study (Ref)	N	Imaging/ N° MET	Type of Lesion	MDT/ Design	Median FU	Endpoints	Outcome
Harrows (11) SABR-COMET Phase II RCT	16/99	Conv. 1–5		Palliative SOC (PSOC) vs. SABR + PSOC	5.7 yrs	P: OS S: PFS Toxicity, QoL,	8-yr OS: HR 0.50 8-yr PFS: HR. 0.45
Ost (13) STOMP Phase II RCT	62	PET-Cho (1–3)	Nodal 55% M1a 16% 1–2 met 78%	Surveillance vs. MDT: SBRT (81%) or S)	3 yrs	P: ADTF	13 vs. 21 mo (HR 0.60; *p* = 0.11)
Phillips (14) ORIOLE Phase II RCT	4	Conv PSMA-PET (1–3)	Nodal alone 58% mean n° lesions 1.6	Surveillance vs. SBRT	19 mo	6 months PFS Median PFS	81% vs. 39% (*p* = 0.005) HR 0.30; *p* = 0.002)
Siva (17) POPSTAR Phase I	33	CT, BS, F-PET 1–3	Nodal 39% 1 lesion: 67%	SBRT (1 × 20 Gy) (ADT 33%)	24 mo	Local-PFS	2-yr L-PFS 93% 2-yr DFS 39% 2-yr ADTF 48%
Glicksman (16) PSMA MRgRT Phase II	74	PSMA-PET-CT/MR 2 lesions	Nodal 34/37 N1 ≤ 3 in 31/37 M1a: 4	SBRT (87%) or Surgery No ADT	41 mo	P: PSA response S: PSA-PFS and ADTF	51% Median 21 months Median 45 months
Hölscher (18) OLI-P Phase II	63	PSMA-PET/MR 1 lesion	1 lesion: 71% Nodal alone 68%	SBRT 77% CRT 50 Gy 23% No ADT	37 mo	P: Treatment-related toxicity S: PSAFS Time to ADT	No grade ≥ 2 tox Median 13.2 mo Median 20.6 mo
Conde Moreno (48) SBRT-SG05 Phase II	67	PET-Cho/MR 1–5	Nodal 57% Non-spinal bone 36% Spinal bone 6%	SBRT and ADT	41 mo	DPFS	Median DPFS 54.2 mo No grade ≥ 3 tox

Abbreviations: N: number of patients; N° MET: number of metastasis allowed; MDT: metastasis-directed therapy; Conv: conventional; PET-Cho: PET-choline; CT: computerized tomography; BS: bone scan; MR: magnetic resonance; MDT: metastasis-directed therapy; SOC: standard of care; SABR: stereotactic ablative radiation therapy; SBRT: stereotactic body radiation therapy; ADT: androgen deprivation therapy; CRT: conventional radiation therapy; yrs: years; mo: months; P: primary; S: secondary; PFS: progression-free survival; QoL: quality of life; ADTF: freedom from ADT; PSAFS: PSA-free survival. DPFS: disease progression-free survival; HR: hazard ratio; Tox: toxicity.

**Table 2 cancers-15-04159-t002:** Summary of ongoing trials on metastasis-directed radiotherapy in metachronous oligometastatic HSPC.

Trial (Type)	n	Patient Characteristics	Design	Primary Outcome	Status/Estimated Completion Date
Diagnosis with PET/TC w/wo WB MRI					
**NCT03304418RROPE (phase II)**	20	Metachronous oligometastatic (≤3) HSPC (PSMA-PET)	Radium-223+ SBRT (16–32.4 Gy/1–6 fx)	Time to ADT	Active, not recruiting/August 2023
**NCT03902951 (phase II)**	28	Metachronous oligometastatic (≤5) M1a,b prostate cancer patients (exclusive of pelvic nodal N1 metastases)	ADT + Abiraterone + Apalutamide + SBRT (1–5 fx)	% of patients achieving PSA < 0.05 ng/mL	Recruiting/January 2025
**NCT04557059 PRIMORDIUM (phase II)**	412	Metachronous oligometastatic pelvic nodes HSPC (PSMA-PET	*Arm A*: pbRT + Pelvic RT +/− SBRT (ns) + ADT *Arm B*: pbRT + Pelvic RT +/− SBRT (ns) + ADT + Apalutamide	PSMA-PET DPFS	Recruiting/January 2028
**NCT03569241 PEACE V-STORM (phase II)**	178	Metachronous nodal oligometastatic HSPC	*Arm A*: ADT + SBRT (ns) *Arm B*: ADT + WPRT + SBRT (ns)	DPFS	Active, not recruiting/April 2025
**NCT04641078 DART (phase II)**	128	Metachronous oligometastatic pelvic nodes (≤5) HSPC (PSMA-PET)	*Arm A*: SBRT (ns) + Darolutamide *Arm B*: SBRT (ns)	DPFS	Recruiting/February 2026
**NCT03795207 POSTCARD (phase II)**	96	Metachronous oligometastatic Pelvic nodes (≤5) HSPC	*Arm A*: SBRT (3 fx) + Durvalumab Arm B: SBRT (3 fx)	2 years PFS	Active, not recruiting/November 2024
**NCT04748042 FAALCON (phase II)**	29	Metachronous oligometastatic (≤5) HSPC (PSMA-PET)	*Arm A*: RT (ns) + Abiraterone + ADT + Olaparib	% without treatment failure at 24 months	Recruiting/May 2025
**NCT05146973 ProstACT TARGET/SATURN (phase II)**	50	Metachronous biochemical & oligometastatic HSPC (PSMA-PET)	*Arm A*: 177Lu-DOTA-TLX951 + RT (ns)	BRFS	Recruiting/June 2025
**NCT05404139 DIRECT (phase II)**	66	Metachronous oligometastatic (≤10) HSPC (PSMA-PET)	*Arm A*: ADT + SBRT (ns) *Arm B*: Enzalutamide + ADT + SBRT (ns)	PFS	Not yet recruiting/March 2026
**NCT04031378 (phase II)**	100	Synchronous/metachronous oligometastatic (≤3) prostate cancer	*Arm A*: SBRT (24 G/1 fx) *Arm B*: SBRT (24 Gy/1 fx) + systemic therapy	BRFS	Unknown
**NCT04599686 (not applicable)**	100	Metachronous oligometastatic (≤3) M1a,b prostate cancer patients (PSMA-PET)	ADT vs. SBRT (30–50 Gy/3–5 fx)	1-year ADT-free survival	Recruiting/October 2025
**NCT05352178 SPARKLE (phase III)**	873	Metachronous oligometastatic pelvic nodes (≤5) HSPC (PSMA-PET)	*Arm A*: MDT *Arm B*: ADT +ADT *Arm C*: MDT + ADT+ Enzalutamide	Poly-metastatic-free survival	Recruiting/April 2032
**NCT03630666 OLIGOPELVIS2 (phase III)**	256	Metachronous oligometastatic pelvic nodes (≤5) HSPC (PSMA-PET)	*Arm A*: Intermittent ADT *Arm B*: Intermittent ADT + IMRT (ns)	PFS	Recruiting/June 2026
**NCT04423211 INDICATE (phase III)**	804	Metachronous biochemical & oligometastatic HSPC	*Arm A*: (PET negative extra-pelvic metastases): ADT + Pelvic RT (ns) *Arm B*: (PET negative extra-pelvic metastases): ADT + Pelvic RT (ns) + Apalutamide *Arm C*: (PET positive extra-pelvic metastases): ADT + Pelvic RT (ns) + Apalutamide *Arm D*: (PET positive extra-pelvic metastases): ADT + Pelvic RT (ns) + Apalutamide + SBRT	PFS	Recruiting/December 2032
**NCT04983095 METRO (phase III)**	114	Synchronous/metachronous oligometastatic (≤3) HSPC (PSMA-PET)	*Arm A*: pRT (ns) + ADT *Arm B*: pRT (ns) + ADT + SBRT	PFS	Recruiting/December 2029
**NCT04115007 PRESTO (phase II)**	350	Synchronous/metachronous oligometastatic (≤5 at least 1 bone/lung mtx) HSPC	*Arm A*: pRT (60 Gy/20 fx) + ADT +/− ST + (SBRT 30 Gy/3 fx) *Arm B*: pRT (60 Gy/20 fx) + ADT +/− ST	Castration resistant prostate cancer-free survival	Recruiting/June 2028
**NCT03940235 RADIOSA (phase II)**	150	Metachronous oligometastatic (≤3) M1a,b prostate cancer patients	SBRT vs. SBRT + ADT	PFS	Recruiting/April 2024
**START-MET (phase III)**	266	Synchronous/metachronous oligometastatic (≤3) with PSMA-PET (≤5) HSPC	*Arm A*: pRT (ns) + ADT + SGADT + SBRT (ns) *Arm B*: pRT (ns) + ADT + SGADT	rPFS	Recruiting/January 2027
**NCT03361735 RA 223 + SBRT (phase II)**	24	Synchronous/metachronous oligometastatic (≤4, at least 1 bone mtx) HSPC	*Arm A*: ADT + Radium-223 + SBRT (3–5 fx)	Time treatment failure & response rate	Recruiting/December 2023
**NCT02759783 CORE (phase II/III)**	245	Metachronous oligometastatic (≤3): Prostate, Breast, NSCLC	SOC vs. SOC + SBRT (ns)	PFS	Active, not recruiting/October 2024
**NCT03043807 (phase II)**	26	Metachronous oligometastatic (≤5) HSPC	Docetaxel + ADT+ RTp+ SBRT (ns)	Efficacy as 3y-BRFS	Active, not recruiting/February 2024
**NCT04037358 RAVENS (phase II)**	64	Metachronous oligometastatic (≤3) HSPC (at least 1 bone mtx)	*Arm A*: SBRT (1–5 fx *Arm B*: Radium-223 + SBRT (1–5 fx)	PFS	Active, not recruiting/December 2025
Diagnosis with conventional image or PET					
**NCT03784755 PLATON (phase III)**	410	Synchronous/metachronous oligometastatic (≤5) HSPC	*Arm A*: SOC + pRT (ns) *Arm B*: SOC + pRT(ns) + SBRT (ns)	PFS	Recruiting/December 2025

Adapted with permission from Ref. [65]. Abbreviations: ADT = androgen deprivation therapy; BRFS = biochemical relapse-free survival; DPFS = distant progression-free survival; mdtfx = fractions; Gy = gray; HSPC = hormone-sensitive prostate cancer; IMRT = intensity-modulated radiotherapy; MDT = metastasis-directed therapy; mtx = metastases; ns = not specified; NSCLC = non–small cell lung cancer; pbRT = prostate bed radiotherapy; PFS = progression-free survival; pRT = prostate radiotherapy; PSA = prostate-specific antigen; PSMA-PET = prostate-specific membrane antigen-positron emission tomography; rPFS = radiographic progression-free survival; RP = radical prostatectomy; RT = radiotherapy; SGADT = second-generation antiandrogen therapy; SBRT = stereotactic body radiation therapy; SOC = standard of care; ST = systemic therapy, WB MRI = whole-body magnetic resonance imaging.

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
