# Peer review of "Node Oligorecurrence in Prostate Cancer: A Challenge"

_cancers, 2023, doi:10.3390/cancers15164159_

Round 1
Reviewer 1 Report
Accept as it
Author Response
We thank the reviewer for his/her time and acceptance as it is.
Reviewer 2 Report
The paper is an interesting review about of the nodal oligorecurence in prostate cancer and metastasis-directed therapy using SBRT as good option to patients and the contrast with other methodologies. The paper is well written, the topic is interesting for the clinical audience and researchers. The authors shown relevant data about several treatments for nodal recurrence, the potential and the possible utility of a new imaging guided SBRT. However some points could be discussed. In the line 179 the authors mentioned somatic mutations in different genes ATM, BRCA1/2,Rb1 and p53, and shows that patients without a high risk mutation who received MDT had superior outcomes. In this point the authors can discuss if they have some data about of the gene expression independently to the mutations. They can find some data in the gene databases. Also in the case of the WNT pathway?
The authors can include some data about gene expression of the signaling cascade in WNT pathway? They could discuss this part.
The use of genomic test associated to molecular imaging guided SBRT would be improve the clinical trials, the authors can discuss this point?
Author Response
We thank the reviewer for the valuable comments and suggestions. As a result, we have engaged in a more in-depth discussion concerning recent developments in the field of biomarkers and genomic profiling in this setting. Additionally, we have incorporated three new references.
Reviewer 3 Report
Zapatero et al present a review of oligometastatic nodal disease in HSPC specifically focusing on studies for SBRT/MDT in the context of novel imaging (PSMA PET). Overall, this is a timely and important topic in prostate cancer management, wherein stage migration and early detection of these nodal metastasis is presenting this problem in clinic on a routine basis. Overall this is well done. Some minor editing of the language to read a bit more smoothly would be suggested.
Specific comments:
1. Line 163 -- this is oddly phrased, be specific as to the implications of EMBARK to oligometastasis management
2. Line 286 -- be specific that the data cited here is for nonmetastatic BCR being treated with salvage radiotherapy. Separately, the standard of care for BCR that is NOT being treated with salvage radiotherapy is either observation or intermittent ADT, and the data is much less robust for supporting ADT without standard radiotherapy. Furthermore, I would suggest being very clear that although ADT is felt to be synergistic with standard fractionated radiotherapy, the same biologic rationale for synergy with SBRT/ablative techniques does not follow.
3. Lastly, given that the natural history of biochemically recurrent prostate cancer is highly variable, and many patients ultimately do not die of prostate cancer with long times to clinical metastasis in some cases, it should be acknolwedged that ultimately the long term benefit of early detection via PSMA PET and aggressive treatment may not benefit patients. For example, if a patient with BCR otherwise had a long PSADT and would not develop overt metastasis for 10 years, but can early nodal metastases can be identified much earlier by PSMA PET, treating those metastasis is not guaranteed to result in benefit. Some discussion is warranted regarding merging the vast data regarding the natural history of BCR after prostatectomy and the unknown long term effects of this upstaging disease with PSMA PETs and subsequent aggressive therapy with SBRT + ADT.
Some editing of flow of the english and specific correction of awkward phrasing is recommended.
Author Response
Specific comments:
1. Line 163 -- this is oddly phrased, be specific as to the implications of EMBARK to oligometastasis management:
Answer (A): According to the reviewer suggestion we have been more explicit in what we think are the potential implications of EMBARK trial (lines 167-169 of the revised manuscript)
2. Line 286 -- be specific that the data cited here is for nonmetastatic BCR being treated with salvage radiotherapy. Separately, the standard of care for BCR that is NOT being treated with salvage radiotherapy is either observation or intermittent ADT, and the data is much less robust for supporting ADT without standard radiotherapy. Furthermore, I would suggest being very clear that although ADT is felt to be synergistic with standard fractionated radiotherapy, the same biologic rationale for synergy with SBRT/ablative techniques does not follow.
(A): We have reformulated this paragraph in the revised manuscript (lines 322-327). Regarding the specifics of radiobiology of SBRT combined with AD, we think that is a topic out of the scope of the present review.
3. Lastly, given that the natural history of biochemically recurrent prostate cancer is highly variable, and many patients ultimately do not die of prostate cancer with long times to clinical metastasis in some cases, it should be acknolwedged that ultimately the long term benefit of early detection via PSMA PET and aggressive treatment may not benefit patients. For example, if a patient with BCR otherwise had a long PSADT and would not develop overt metastasis for 10 years, but can early nodal metastases can be identified much earlier by PSMA PET, treating those metastasis is not guaranteed to result in benefit. Some discussion is warranted regarding merging the vast data regarding the natural history of BCR after prostatectomy and the unknown long term effects of this upstaging disease with PSMA PETs and subsequent aggressive therapy with SBRT + ADT.
(A) The impact of molecular imaging-guided treatment on clinical outcomes is a very interesting subject that has been addressed through the manuscript (lines 128-129, 152-154, and 238-240 of the revised manuscript). We completely agree with the referee that patients with indolent or low risk biochemical failure following radical prostatectomy may no benefit from this approach and have included an specific comment for clarity purpose (lines 242-244).